# Axin2/Conductin Is Required for Normal Haematopoiesis and T Lymphopoiesis

**DOI:** 10.3390/cells11172679

**Published:** 2022-08-28

**Authors:** Jolanda J. D. de Roo, Amiet Chhatta, Laura Garcia-Perez, Brigitta A. E. Naber, Sandra A. Vloemans, Daniela C. F. Salvatori, Karin Pike-Overzet, Harald Mikkers, Frank J. T. Staal

**Affiliations:** 1Department of Immunology, Leiden University Medical Center, 2333 ZA Leiden, The Netherlands; 2Department of Clinical Sciences, Anatomy and Physiology Division, Faculty of Veterinary Medicine, Utrecht University, 3584 Utrecht, The Netherlands; 3Department of Cell & Chemical Biology, Leiden University Medical Center, 2333 ZA Leiden, The Netherlands

**Keywords:** WNT- thymus-hematopoietic stem cell

## Abstract

The development of T lymphocytes in the thymus and their stem cell precursors in the bone marrow is controlled by Wnt signaling in strictly regulated, cell-type specific dosages. In this study, we investigated levels of canonical Wnt signaling during hematopoiesis and T cell development within the Axin2-mTurquoise2 reporter. We demonstrate active Wnt signaling in hematopoietic stem cells (HSCs) and early thymocytes, but also in more mature thymic subsets and peripheral T lymphocytes. Thymic epithelial cells displayed particularly high Wnt signaling, suggesting an interesting crosstalk between thymocytes and thymic epithelial cells (TECs). Additionally, reporter mice allowed us to investigate the loss of Axin2 function, demonstrating decreased HSC repopulation upon transplantation and the partial arrest of early thymocyte development in Axin2^Tg/Tg^ full mutant mice. Mechanistically, loss of *Axin2* leads to supraphysiological Wnt levels that disrupt HSC differentiation and thymocyte development.

## 1. Introduction

A number of evolutionarily conserved pathways regulate the development and maintenance of adult stem cells, including the Notch, Bone Morphogenetic Protein (BMP), Hedgehog, Fibroblast Growth Factor (FGF), Transforming growth factor beta (TGF-β) and Wnt signaling pathways [1]. These pathways maintain a small subset of the total cell pool containing stem cells in a specific microenvironment—a niche—that are crucial for homeostasis or renewal after injury. Among these pathways, the Wnt pathway has been well studied and shown to be absolutely required for the self-renewal of many types of adult stem cells [2,3]. Compared with the convincing studies on the role of Wnt signaling in adult stem cells in the skin and the gut, a role for Wnt in adult hematopoietic stem cells (HSCs) has proven much more difficult to demonstrate [4]. In studies reporting an important role for Wnt signaling in blood cells, Wnt was shown to be required for normal HSC self-renewal and therefore for efficient reconstitution after transplantation [5]. Additionally, Wnts are crucial for the development of T cells in the thymus [6,7], as blocking of Wnt signaling leads to the near-complete arrest of the early stages of T cell development [8]. Conversely, strong activation of the Wnt pathway also leads to developmental abnormalities due to loss of stemness and increased differentiation [5,9]. These findings led to the concept that cell-type-specific, “just-right” levels of Wnt signaling are required for proper development and stem cell function. Indeed, this dosage model of Wnt signaling helps to explain the seemingly contradictory reports on Wnt signaling in haematopoiesis and lymphopoiesis [10].

In the most commonly studied Wnt pathway (canonical Wnt Signaling), all 19 Wnt proteins are able to bind to a receptor complex composed of a member of the 10 Frizzled transmembrane receptors and an LRP5 or LRP6 co-receptor. Without Wnt binding to this complex ligand, the cytoplasmic signaling molecule β-catenin is constantly degraded in a large protein complex. This complex is composed of two negative regulatory kinases, including Glycogen Synthase Kinase 3 beta (GSK-3β) and Casein kinase 2 (CSK2), and at least two anchor proteins that also function as tumor suppressor proteins, namely, Axin1 or Axin2 and APC (adenomatous polyposis coli). APC and Axin function as negative regulators of the pathway by sequestering β-catenin in the cytoplasm. Activation of the pathway by Wnt leads to inactivation of the destruction complex, allowing a buildup of β-catenin and its migration to the nucleus. In the nucleus, β-catenin binds to members of the TCF/LEF transcription factor family, thereby converting them from transcriptional repressors into transcriptional activators [10,11,12]. Axin2 is usually considered to be redundant to Axin1 and is a direct target gene of the pathway [13]. Therefore, it is often measured as a canonical Wnt signaling reporter protein. Axin2 knock-out mice reportedly have few abnormalities, those they do exhibit being mostly in tooth and skull development [14].

A problem encountered when studying Wnt signaling in blood and immune cells is the lack of proper tools with which to study the activity of the pathway in intact blood cells, as in most other organ systems histological or molecular tools are used rather than flow cytometry. Since flow cytometry is the most commonly used technique for studying immune cells, we previously used the FACS-Gal assay to assess Wnt signaling in the Axin2^LacZ^ reporter mouse—the mouse model that is considered to most faithfully report Wnt activity [5,15]. However, the FDG assay requires hypo-osmotic loading of the β-galactosidase substrate, leading to altered scatter properties and sometimes to the apoptosis of cells. In addition, the substrate requirements when using enzymatic reporters can be overcome by using auto-fluorescent proteins, such as green fluorescent protein (GFP), whose expression is controlled by the signaling pathway under study. Hence, we developed a mTurquoise2 fluorescent-protein-based canonical Wnt reporter utilizing CRISPR-Cas9 to generate a similar Axin2 reporter with mTurquoise2 instead of LacZ [16]. Here, we describe a detailed phenotypic and functional analysis of the blood and immune cells and organs of these mice. These analyses prompted us to study canonical Wnt signaling upon transplantation of HSCs into recipient mice. Unexpectedly, we discovered that mice in which both *Axin2* alleles were targeted with the mTurquoise2 gene and thereby rendered *Axin2* knock-out null mutants showed defects in hematopoiesis and T cell development. Hence, our data show that Axin2 plays a non-redundant role in regulating HSC differentiation and early T cell development.

## 2. Materials and Methods

### 2.1. Mice and Cell Collection

All mouse procedures were approved by the national commission for laboratory animal experiments, Centrale Commissie Dierproeven (CCD). Male and female wild-type, heterozygous (Tg/wt) and homozygous (Tg/Tg) Axin2^em1Fstl^ mice (now available from the Jackson laboratory) have been described before [16]. Wildtype *C57Bl/6 Ly5.1* mice and Rag1^−/−^ mice, 6–12 weeks old, were used as donors and recipients for the transplantation experiments.

Ilea, femurs and sterna were harvested and crushed for bone marrow collection. Fresh or thawed bone marrow was resuspended in cold IMDM medium (Thermo Fisher Scientific (Gibco), Waltham, MA, USA), 2.5% FCS (Greiner Bio-one B.V. Alphen aan den Rijn, The Netherlands), supplemented with 100 U/mL penicillin and 100 μg/mL streptomycin (Thermo Fisher Scientific).

For the competitive transplantation, mouse Lin^−^ cells were purified by cell sorting on a FACSAriaII cell sorter (BD Biosciences, Franklin Lakes, NJ, USA) with the following anti-mouse antibodies: CD3-biotin, Ter119-biotin, GR-1-biotin, B220-biotin, CD11b-biotin, cKit-APC and Sca1-Pe-Cy7, then counterstained with Streptavidin-PE (all from eBiosciences, San Diego, CA, USA). Finally, sorted cells were collected into cold StemSpan serum-free expansion medium (StemCell Technologies Inc., Vancouver, BC, Canada) with 100 U/mL Penicillin and 100 μg/mL streptomycin (Thermo Fisher Scientific). Purity was always >95% upon re-analysis. 

### 2.2. Competitive Transplantation

Competitive transplantation was performed using the Ly5.1/Ly5.2 congenic system. *C57Bl/6 Ly5.1* mice (aged 10 weeks) were lethally irradiated with 8.08 Gy X-rays using orthovoltage irradiation and transplanted 24 h later via tail vein injection with test (Ly5.2, Axin2-mTurquoise2) and competitor (Ly5.1, wild type) cells, as indicated. The mice were fed with Diet gel recovery (Clear H_2_O, Portland, MA, USA) and triple antibiotic water until sacrifice. The recipient mice were transplanted with 1 × 10^5^ Ly5.1 and 1 × 10^5^ Ly5.2 Lin^−^ cells, together with 5 × 10^5^ Ly5.1 spleen support cells. At 6 weeks after transplantation, the mice were sacrificed by O_2_/CO_2_ inhalation, and bone marrow, spleens and thymuses were isolated. Bone marrow, spleen and thymus cell suspensions were obtained through crushing and homogenization by passage through a 70 μm filter. All cell suspensions were suspended in cold IMDM medium (Gibco, Life Technologies, Bleiswijk, The Netherlands), 2.5% FCS (Greiner Bio-one B.V. Alphen aan den Rijn, The Netherlands), supplemented with 100 U/mL penicillin and 100 μg/mL streptomycin (Gibco, Life Technologies).

### 2.3. Measuring Wnt Signaling in Thymocytes

Thymocytes were stained with the following antibodies: Ter119-biotin, GR-1-biotin, B220-biotin, NK1.1-biotin, Streptavidin-PE-Cy7, CD3-APC, TCRβ-FITC (BD Biosciences, Franklin Lakes, NJ, USA), CD11b-biotin, CD8-PerCP (Biolegend, San Diego, CA, USA), CD45.2-APC-eFluor780, CD4-BV650 (eBioscience, Santa Clara, CA, USA). The biotinylated antibodies were used to exclude lineages other than the T cell lineages. Cells were stained in FACS buffer (PBS, 2% BSA, 0.1% sodium azide) and subsequently measured by flow cytometry.

### 2.4. Histology

Steady-state Axin2-mTurquoise mice were sacrificed at the age of 12 weeks for histological analysis. Small and large intestine or thymus tissues were immerged in 4% Paraformaldehyde in Phosphate-Buffered (PFA) solution at RT. After 1 h, the pieces were transferred from 4% PFA to 15% Sucrose-PBS solution for a minimum of 2 h and subsequently to 30% Sucrose-PBS solution O/N at 4 °C. Finally, the pieces were embedded in Tissue-Tek^®^ OCT compound (Sakura^®^ Finetek, Radnor, PA, USA) for further analysis. Subsequently, 5 μm frozen sections were processed for hematoxylin and eosin (H&E) staining according to standard procedures and for endogenously expressed mTurquoise2 fluorescence analysis. For the latter, sections were dried for 1 h at RT. DNA counterstaining was performed with TO-PRO-3 (1:1000, Thermo fisher). Images were taken using a Leica TCS SP5 confocal microscope.

### 2.5. Thymocyte Cytospinning and Immunostaining

Steady-state Axin2-mTurquoise2 thymocyte cytospinning and immunostaining against total β-catenin, as well as confocal imaging, were performed as described in [17]. Two to three sections per cytospin were imaged and analyzed using the open-source image-processing software Fiji/ImageJ [18]. Calculations of the cytoplasmic–nuclear β-catenin ratios were performed for the same cells and with background-corrected fluorescent values, as described in [17]. Figure images were created with the open-source QuickFigures ImageJ plugIn [19].

### 2.6. Flow Cytometry Settings and Analyses

For the flow cytometry analyses, cells were stained with monoclonal antibodies against the following anti-mouse molecules: CD3-biotin, Ter119-biotin, GR-1-biotin, B220-biotin, CD11b-biotin, NK1.1-biotin, CD3-biotin, CD4-biotin, CD8-biotin, cKit-BV650, Sca1-PE, CD135-PerCP, CD34-APC, CD25-PE, CD44-APC, CD3-APC, CD8-PerCP, CD-BV650, CD45.1/Ly5.1-PE-Cy7, CD45.2/Ly5.2-APC-Cy7 (all from eBiosciences, CA, USA). For secondary detection of biotinylated antibodies, streptavidin conjugated with FITC or PE-Cy7 was used (eBiosciences, San Diego, CA, USA). All flow cytometry measurements (Canto II BD biosciences or LSRII BS Biosciences) were calibrated first with BD™ CompBead Plus, κ/Negative Control (BSA) Compensation Plus (7.5 µm). Recommended flow cytometry settings are specified in [17]. Flow cytometry analyses were performed using FlowJo software (Treestar, Ashland, OR, USA). 

### 2.7. Statistical Analysis

Statistical analysis was determined by one-way ANOVA or two-way ANOVA followed by the Kruskal–Wallis multiple comparisons test or the Mann–Whitney U test, as indicated in the figure legends. All analyses were performed using GraphPad Prism version 9.0.1 for Windows (GraphPad Software, San Diego, CA, USA, www.graphpad.com; accessed on 1 January 2022).

## 3. Results

### 3.1. Canonical Wnt Activity in the Thymus

Previously, we reported the generation of a new Wnt reporter strain, Axin2^em1Fstl^, in which the GFP variant mTurquoise2 was targeted into one of the alleles of the Wnt target gene Axin2 [16] (Figure 1A). In order to determine reporter expression strength in the hematopoietic system, as well as possible *Axin2* mutant effects, we measured canonical Wnt expression in the thymus as a reference Wnt organ in steady-state Axin2-mTurquoise2 mice. The Axin2-mTurquoise2 heterozygote (Tg/wt) and *Axin2* null mutant (Tg/Tg) showed gradual reductions in total thymocyte cellularity, although no significant effects were visible (Figure 1B), while they showed significantly strong upregulation of mTurquoise2 expression (Figure 1C). In other hematopoietic organs, such as bone marrow and the spleen, a similar decrease in total cellularity simultaneous with *Axin2* allele loss was apparent (Appendix A). 

While the canonical Wnt activity of thymocytes has been documented in detail [20], it has been much more difficult to assess Wnt activity within the stromal compartment, in part because thymic epithelial cells (TECs) are heterogeneous and thus hard to isolate and characterize functionally and molecularly [21]. Although thymic stromal cells are known to express the Wnt ligands Wnt-4, Wnt-7a and -7b, and Wnt-10a and -10b [22], these cells also undergo canonical Wnt signaling themselves [23]. However, the mechanism by which active canonical Wnt signaling is regulated in the adult thymus has remained elusive due to its low expression. We measured mTurquoise2 expression in thymus cryosections from wild-type (wt) littermates and heterozygous (Tg/wt) and homozygous (Tg/Tg) mice. In our model, dim expression of Wnt activity was also detected in the heterozygous (Tg/wt) and homozygous (Tg/Tg) thymus samples (Figure 1D, lower row), which is reflected in the shades of LUT colors ranging from dark purple (low expression) to bright yellow (high expression). However, the clear increment in mTurquoise2 expression, as shown by flow cytometry (Figure 1C), was hard to distinguish from the background signal in the wild type (wt) when analyzing the cryosections by confocal microscopy, while in the ileum, which was supplemented as a canonical signaling reference organ, this stepwise increment in overall mTurqoise2 expression was more readily visible (Figure 1D, upper row). Nonetheless, it remains difficult to determine canonical Wnt signaling via this technique, which suggests that a cell-based approach, such as will be described later in this report, is more suitable. Finally, there was no visual indication of an abnormal thymic structure, even though we report a minor reduction in total thymocyte cellularity (Figure 1B). 

### 3.2. Wnt Activity under Steady-State Conditions

In order to determine the in vivo canonical Wnt signaling status within the hematopoietic system, we analyzed blood and immune cells using flow cytometry for the expression of mTurquoise2. As we reported previously when using Axin2/conductin^LacZ/+^ [5], most blood cell populations showed some canonical Wnt activity during development; however, T cells are the only mature immune/blood cells which have discernible Wnt activity. In line with previous data, canonical Wnt signaling active cells are present in Lin^−^Sca1^+^c-Kit^+^ (LSK) stem cells; however, essentially, the short-term (ST) HSC population expressed noticeable reporter expression (Figure 2A,C). Nonetheless, canonical Wnt signaling was substantially more active in thymocytes (Figure 2B). Similar to the Axin2^LacZ^ mice, the heterozygous mice (Tg/wt) clearly displayed canonical Wnt active cells from the CD4^−^CD8^−^ double-negative (DN)1 population onwards; however, the more mature thymocytes, such as the CD4^+^CD8^+^ double-positive (DP) and single-positive (SP)8 thymocytes, demonstrated the highest frequencies of mTurquoise2-expressing cells (Figure 2B). Interestingly, median mTurquoise2 expression within the thymic lineage-specific subsets was lower in the later stages of thymocyte development, indicating that canonical Wnt signaling is regulated within an optimal range (Figure 2D). In fact, ETPs expressed the lowest reporter activity strength in T cell development. This is similar to the results we previously reported for the Axin2^LacZ^ reporter [5], although the expression was lower. Most likely, LacZ reporter signal detection in rare ETPs was an artifact of the hypo-osmotic β-galactosidase staining procedure, which is known to lead to cell death in highly apoptosis-sensitive cells [24]. 

Although the heterozygous genotype (Tg/wt) is the genotype that is typically used to analyze Wnt signaling levels, we also checked double transgenic Tg/Tg mice that were in fact null mutants for *Axin2*. The expression of lineage-specific subsets was similar to the expression of wild-type (wt) and heterozygous (Tg/wt) mice, indicating that no effect was visible related to the loss of Axin2 (data not shown). In the double-transgenic mice, the frequencies of reporter-expressing cells were generally higher than in the heterozygous mice, although reporter gene expression variation was noticeable in the early stem cell populations due to low cell numbers. Indeed, in the thymocyte subsets we often found frequencies of mTurquoise2-expressing cells that were more than two times higher than those found in the heterozygotes (Figure 2A,B). This suggests that loss of Axin2 resulted in significantly higher frequencies of Wnt active cells compared to cells in which Axin2 was still expressed due to the intact allele. Strikingly, the ETP subpopulation of the *Axin2* null mutants (Tg/Tg) showed a high frequency of Wnt-expressing cells, with high median mTurquoise2 expression (Figure 2B,F). 

### 3.3. Axin2 Is Required for Normal Post-Transplantation Hematopoiesis and T Cell Development 

As Wnt signaling has been implicated in the self-renewal of HSCs, it was of interest to investigate Wnt signaling levels after the transplantation of stem cells into irradiated recipient wild-type mice. In order to carefully analyze the fate of transplanted cells and their progeny, competitive transplantation is often used; a minor allelic variant of CD45 (also referred to as Ly5) was used to discriminate wild-type (Ly5.1) competitors from the test population (Axin2-mTurquoise2 Ly5.2 cells). 

Detailed analyses of Ly5.2 versus Ly5.1 cells in various subsets revealed that mice transplanted with Ly5.1 and Axin2-mTurquoise2 heterozygous stem cells (Tg/wt) or Axin2-mTurquoise2 homozygous stem cells (Tg/Tg) showed overall similar distributions of test (ly5.2) versus competitor (ly5.1) cells in all HSC and progenitor subpopulations in the bone marrow (Figure 3A). The heterozygous stem cells (Tg/wt) clearly outcompeted the Ly5.1 stem cells, which pattern was maintained throughout all the differentiation subsets. Contrarily, the homozygous stem cells (Tg/Tg) were outcompeted by the wt Ly5.1 stem cells and sometimes displayed a five-fold lower output (Appendix A). This was especially apparent in the LT-HSC, ST-HSC and MPP subsets, but, nevertheless, other progenitor populations were affected as well. The lower number of cells developed from homozygous Tg/Tg HSCs was also reflected in increased cell numbers for the total Ly5.1 compartment, indicating that stress on the stem cell compartment may have provoked signaling, leading to increases in Ly5.1 cells (Appendix A).

In the thymus, a similar phenomenon was observed throughout almost all T cell developmental stages. The developmental block for the homozygous test cells (Tg/Tg) was most prominent in the early T cell stages (Figure 3B). Of interest, similar to the bone marrow, the heterozygous cells (Tg/wt) outcompeted the wt Ly5.1 competitor stem cells, except for the DN1 subset, in which both the wt Ly5.1 competitor and the Ly5.2 test cells contributed more or less equally to the population (Appendix A). Later T cell development subsets were less affected, but still an effect of Axin2 deletion could be documented (data not shown).

When comparing the frequencies of mTurquoise2-expressing cells, we observed that a slight increase in the numbers of Wnt active cells in the LSK compartment in both heterozygous and homozygous mice was primarily driven by the LT and ST HSCs, although the variation between mice was high (Figure 4A). In turn, the multipotent progenitors (MPPs) and the common lymphoid progenitors (CLPs) seemed to be the least responsive in terms of frequency of mTurquoise2-expressing cells, whereas the myeloid progenitors (MPs) showed the highest increase in Wnt active cells after transplantation. In the thymus, on the other hand, almost all T cell subpopulations demonstrated increased mTurquoise2-positive frequencies compared to the steady-state mice. However, the ETPs from the homozygous mice (Tg/Tg) constituted the only T cell population which showed a decrease in Wnt active cells compared to the steady-state homozygous mice (±45% versus ±20% mTurquoise2 cells) (Figure 2B and Figure 4B). Of particular interest is that the mice with very low contributions of test cells (ly5.2) were also those which did not express detectable Wnt signaling. 

Indeed, in the heterozygous reporters, we observed about 20–50% higher median mTurquoise2 expression levels than under steady-state conditions (Figure 2C and Figure 4C), both in the bone marrow total LSK and hematopoietic progenitor stages, whereas in the early HSC stages and the T cell stages in the thymus we observed larger increases and more dichotomous mTurquoise2 expression (Figure 4C,D). LT HSCs and ETPs, especially, showed increased levels of Wnt activity. Instead, the levels of Wnt signaling in the double-transgenic mice were generally higher than in the heterozygous reporter mice. Early HSC stages, in particular, showed a dichotomous mTurquoise2 expression ranging between no expression to very high median reporter expression (Figure 4E). Interestingly, in the T cell developmental stages in the thymus, the ETPs seemed to reach a maximum Wnt activity state and SP8 cells (which incidentally also proliferate faster than CD4 SPs) reached the highest Wnt activation of all the subsets. When comparing the steady-state homozygous reporter expression (Figure 2F) with the transplanted homozygous cells, only the DN4 cells were found to react with slightly increased median reporter expression (Figure 4F), suggesting that this decision point requires stricter Wnt signaling regulation. 

### 3.4. Wnt Signaling Activity Is Increased in Axin2 Mutants

The data presented in Figure 1 suggested that higher than normal Wnt signaling levels may occur as a result of the lack of Axin2. However, previous studies on the BATgal canonical Wnt signaling reporter claimed that nuclear signaling within the thymus is low [25]. We also showed that cryosection mTurquoise2 reporter analyses are difficult to interpret due to high background signals that limit dim Axin2-mTurquoise2 signal detection. We therefore sought to measure Wnt signaling activity on a per cell basis. As *Axin2*, which is the most commonly measured Wnt target gene, could not be measured here to support our reporter activity data, we used another hallmark of Wnt signaling, namely, increased nuclear β-catenin expression. β-catenin is also known to exert Wnt-independent functions [26] and can be regulated by multiple proteins. We opted to measure the localization of total β-catenin in the cytoplasm and the nucleus [17]. Figure 5A shows the dim expression of β-catenin and mTurquoise2 co-localization; however, the expression of mTurquoise2 seems to be more concentrated outside the nucleus (TO-PRO-3 staining). The low cytoplasmic-to-nuclear ratio hindered the detection of cytoplasmic mTurquoise2 or β-catenin, for which precise per cell analysis is required. To demonstrate the complex cellular function of β-catenin, we studied the ratio of nuclear β-catenin (active canonical Wnt signaling) to cytosolic β-catenin. As Axin2 forms part of the destruction complex that plays an important role in targeting cytoplasmic β-catenin to proteasomal degradation, the absence or disruption of Axin2 protein could cause an accumulation of either nuclear and/or cytoplasmic β-catenin. This approach was chosen to fairly assess the possible impact of *Axin2* disruption on β-catenin regulation. Indeed, a gradual increase in the nuclear-to-cytoplasmic ratio of β-catenin was recorded in pan-thymocytes (Figure 5B), with increased canonical Wnt signaling being attributed to the loss of Axin2. 

## 4. Discussion

In this study, we made use of an Axin2-mTurqoise2 reporter mouse model to detect canonical Wnt signaling in steady-state and stress hematopoiesis. As the homozygous reporter mice are a full mutant for *Axin2*, we could as well study the lack of Axin2 in the same cellular subsets. A similar differential optimum of Wnt signaling was observed to those observed in previous reporters [5,27,28]; however, the Axin2-mTurquoise2 reporter simplifies the detection of the “just right” model of Wnt signaling. Moreover, hematopoiesis and T lymphopoiesis were found to be differentially affected by increased canonical Wnt signaling activity. 

Canonical Wnt signaling has been reported to be important for thymic organogenesis and T lymphocyte differentiation/maturation [8]. Both TECs and thymocytes express Wnt ligands and influence each other through Wnt activation at distinct maturational stages to establish the complex intrathymic expression patterns [23]. We measured a particular Wnt activation pattern in the Axin2-mTurquoise2 mice which was most notably altered at the ETP stage in the *Axin2* null mutants (Tg/Tg). This suggested that, even though ETPs have the lowest Wnt activity during T cell development, they are capable of higher cell signaling levels when Axin2 is dysfunctional. Brunk et al. demonstrated that TEC Wnt activation is primarily driven by autocrine Wnt ligands and that T cell development is not directly dependent on TEC-provided Wnt ligands [29]. This could explain why the lack of Axin2 did not affect the subset composition of the T cell pool, although a small decrease in total thymocyte frequency was observed. Interestingly, the β-catenin/TCF pathway was shown to be an important regulator of FOXN1 expression in TECs [23]. Nevertheless, overexpressed canonical Wnt signaling by stabilized β-catenin reduced *FOXN1* expression, resulting in failed T lymphopoiesis and T cell hypoplasia [7,30,31]. Therefore, canonical Wnt signaling is probably relatively lowly expressed in the thymic epithelium, as was reported for BATgal reporter mice [25]. In our Axin2-mTurquois2 reporter model, both heterozygous and homozygous, we have shown normal T lymphopoiesis and a normal thymic architecture, suggesting that even though Axin2 deficiency increased nuclear β-catenin expression, the extent of Wnt signaling activation was modest and did not impede normal thymic organogenesis nor T cell development. Furthermore, similar to Axin2^LacZ^ and AXCT2;mTmG reporter mice, we have reported Axin2-expressing cells in the thymus. Interestingly, a role for Axin2 was identified in the characterization of long-lived cortical TEC progenitors [32]. Although the function of these cells remains ambiguous, the Axin2-mTurquoise2 reporter model could be of interest in studying the intricate regulatory behavior of Axin2. Presumably, an anti-GFP secondary antibody will be helpful to ensure better mTurquoise2 detection in thymus cryosections [33]. 

We have reported previously, using combinations of different *Apc*-mutant mouse strains, that hematopoiesis and T lymphopoiesis are regulated by proper Wnt dosages [5]. More specifically, it has been shown that slightly higher Wnt activity levels lead to better reconstitution and T cell development, while levels more than three-fold higher than normal can be detrimental. We demonstrated that this is due to enhanced differentiation and loss of stemness [9]. The results of this study are consistent with the “just right” model of Wnt signaling. Loss of Axin2 leads to supraphysiological Wnt levels, meaning they are too high to support normal development. However, mildly increased Wnt levels in the heterozygous mice led to slightly better T cell development in the early DN stages and in the most immature HSC subsets.

In the T cell compartment, especially, several interesting observations were made regarding mTurquoise2 expression. While all Axin2 heterozygous T cell subsets showed a competitive advantage over wild-type cells after transplantation, the DN1 stage made an equal contribution to the wild-type cells. Interestingly, this is the T cell developmental stage (including the ETPs) in which no clear effect of stabilized β-catenin expression has been documented before [20]. Previously reported roles for canonical Wnt signaling have been related to T cell gate-keeping and T cell fidelity functions in collaboration with Notch signaling [34,35], which explains why no cellular frequency differences were observable in our experiment. Regardless, canonical Wnt signaling plays an important role in thymocyte positive selection. Yu et al. studied transgenic mice that expressed stabilized β-catenin via an Lck-promoter (DN3 onwards) and showed accelerated SP8 thymocyte generation [36,37]. Contrarily, another study with a different strategy for stabilizing β-catenin showed that increased Wnt signaling led to enhanced negative selection [38]. A possible explanation is related to subtle changes in active β-catenin, which could have led to differential outcomes [20]. Nevertheless, when relating these experiments to our Axin2-mTurquoise2 model [39], it seems that the attained Wnt signaling activity ranged somewhere in between.

Axin2, also known as Axil or Conductin, is a negative regulator of the canonical Wnt pathway and functions in the destruction complex as an anchor protein to prevent nuclear translocation of β-catenin [40]. It is thought to modulate the nature or duration of the signaling response, as Axin2 stability fluctuations have drastic effects on signaling [41]. β-catenin, on the other hand, is continually synthesized and rapidly degraded, which makes it practically constitutively present in the cytoplasm. Upon accumulation and stabilization, β-catenin translocates to the nucleus to activate transcriptional activation. This “futile” cycle is especially useful for low-concentration and limited-affinity systems, such as canonical Wnt signaling [42,43]. Therefore, pathway fine-tuning and dosage models are interesting concepts that require intricate measurements of the all interacting proteins. In this study, we have shown differential Axin2 expression and therefore active Wnt signaling activity along hematopoietic and T lymphopoietic development. The conflicting results for the degradation of the β-catenin destruction complex have suggested multiple models of inhibition upon Wnt/β-catenin pathway activation [42,43]. One of them is the disruption of the complex by the dissociation of the Axin subunit; however, the detailed mechanism remains controversial. Axin1 and Axin2 are thought to be equivalent suppressors of canonical Wnt signaling, including Axin1- and Axin2-mediated β-catenin degradation; however, they are not fully redundant. For example, Axin2 was shown to be more effective in the turnover of β-catenin than Axin1 [44]. However, understanding in vivo Wnt pathway kinetics has not been easy due to gene redundancy and their dynamic features. While loss of *Axin1* causes early embryonic lethality [45], *Axin2*-null mice are viable, with craniofacial defects [14,46,47]. Interestingly, knock-in of Axin2 in Axin1-deficient mice fully compensated for the lack of Axin1, indicating that the two genes are functionally equivalent in vivo [48]. The differences, therefore, may be consequences of variations in expression patterns, as Axin1 is ubiquitously expressed, while Axin2 has specific tissue and developmental stage expression. Moreover, recent work has demonstrated that the sequence differences between the scaffolding proteins seem to have biochemical consequences as well [49]. As only Axin2, and not Axin1, is a canonical Wnt target gene, providing a negative regulatory feedback loop, the role of Axin2 deficiency is particularly difficult to study. Hence, we have studied Axin2 deficiency through the Axin2-mTurquoise2 homozygous reporter model, while still being able to utilize its Wnt pathway activity indicator function.

These findings do not indicate that Axin2 expression is not suitable for reporting canonical Wnt signaling. As we have reviewed elsewhere [50], the transcriptional response to Wnt signaling is the ultimate consequence of the signaling cascade that sets a biological response in motion [4]. Therefore, while levels of total β-catenin, or, even better, dephosphorylated (so-called activated) β-catenin, or multimerized binding sites of the TCF/LEF transcription factors linked to reporters, provide valuable bona fide readouts, quantitative measurement of target gene transcription may be most reliable. Amongst the many Wnt target genes, *Axin2* is considered one of the most reliable, as it is not much influenced by the action of other signaling pathways [13]. Hence, a readily measurable in vivo reporter for Axin2, such as we have described here, is a useful tool for studying Wnt signaling.

Collectively, our results suggest a nonredundant role for Axin2 in hematopoiesis. This is an interesting finding, as Axin1 is expressed in HSCs and thymocytes as well. It suggests that, while being co-expressed, Axin2 is required to fine-tune Wnt activity to the “just right” levels that cannot be maintained by Axin1 alone [49], especially in those cell types that require Wnt signaling for their differentiation or proliferation.

## 5. Conclusions

Using a recently developed Wnt signaling reporter mouse, the Axin2 knock-in with Turquoise as a fluorescent marker, we observed a surprising non-redundant role for Axin2 in hematopoiesis and T cell development. In vivo and in vitro studies indicated that Axin2 KO cells have higher Wnt signaling levels than normal. Consistent with the “just right” model of Wnt signaling, this supraphysiological Wnt signaling activity can hamper HSC function, leading to increased myeloid differentiation and decreased T lymphopoiesis.

## Figures and Tables

**Figure 1 cells-11-02679-f001:**
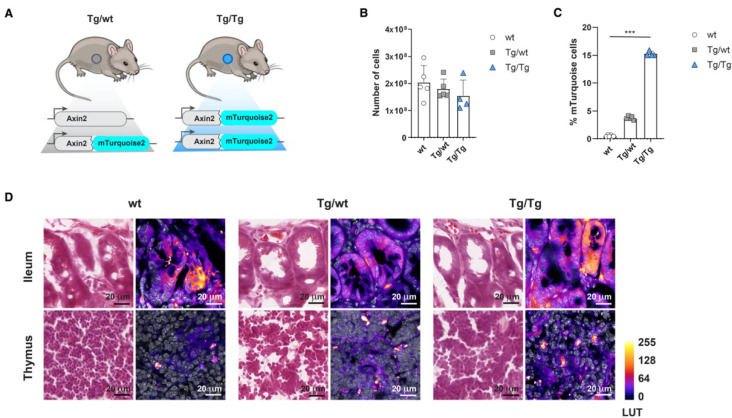
Axin2-mTurquoise2 reporter expression in the thymus and intestine. Axin2-mTurquoise2 reporter expression through FACS and cryosection confocal analysis in the canonical Wnt organs, the thymus and the ileum of the intestine. (**A**) Schematic representation of the Axin2-mTurquoise2 reporter mice, in which mTurquoise2 was targeted into one (Tg/wt) or both (Tg/Tg) of the alleles of the Wnt target gene *Axin2*. Tg/Tg mice in which both *Axin2* alleles were targeted with the mTurquoise2 gene were thereby rendered *Axin2* knock-out null mutants. (**B**) Total thymocyte cell numbers obtained from wild-type (wt), heterozygous (Tg/wt) and homozygous (Tg/Tg) Axin2-mTurquoise2 steady-state mice. Data represent results for five wild-type control mice (wt), five heterozygous Axin2-mTurquoise2 mice (Tg/wt) and four homozygous Axin2-mTurquoise2 mice (Tg/Tg) from two independent experiments. Error bars represent standard errors of the means (SEMs). (**C**) Quantification of the frequencies of mTurquise2 from total thymocytes of Axin2-mTurquoise2 mice. Data represent results for five wild-type control mice (wt), five heterozygous Axin2-mTurquoise2 mice (Tg/wt) and four homozygous Axin2-mTurquoise2 mice (Tg/Tg) from two independent experiments. Error bars represent SEMs. *** *p* < 0.001 (ANOVA test). (**D**) H&E staining (first column of each genotype) and immunofluorescence analysis (second column of each genotype) of mTurquoise2 activity in Axin2-mTurquoise2 thymuses and ilea. Confocal images were captured with a Leica SP5, using identical settings, and processed in ImageJ. Nuclear staining with TO-PRO-3 is shown in grayscale, while differences in mTurquoise2 activity correspond to Fire LUT intensity. Imaged at 100×; scalebars are set at 20 µm.

**Figure 2 cells-11-02679-f002:**
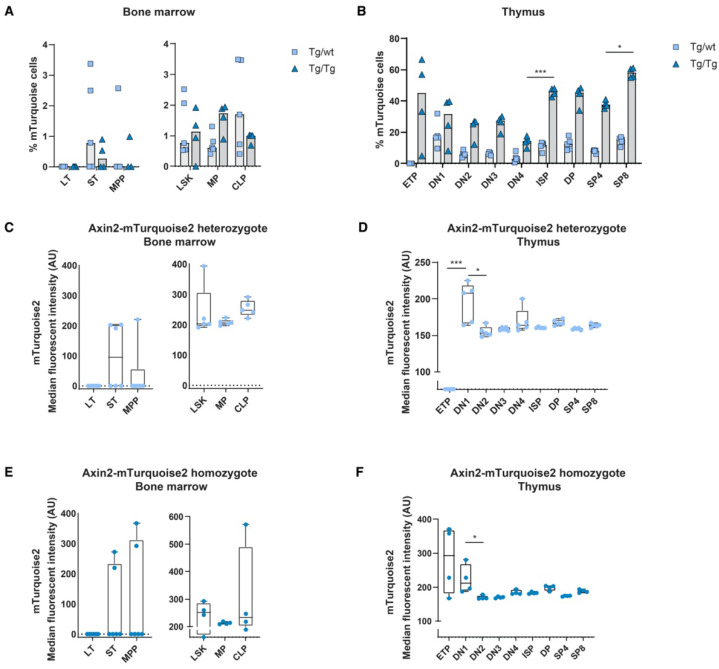
In vivo measurement of canonical Wnt signaling activity throughout hematopoietic development. Activation of the canonical Wnt signaling pathway was measured by FACS using steady-state Axin2-mTurquoise2 reporter mice. Quantification of the frequency of mTurquoise2 (**A**,**B**) and the median fluorescence intensity (MFI) of the mTurquoise2 populations of heterozygous Axin2-mTurquoise2 (Tg/wt) (**C**,**D**) and homozygous Axin2-mTurquoise2 mice (Tg/Tg) (**E**,**F**) for each subset in the bone marrow (**A**,**C**,**E**) and in the thymus (**B**,**D**,**F**). Littermate mice not carrying the reporter transgene (Axin2-mTurquoise2 wild type) were used to define the mTurquoise2 negative population. Data represent results for five heterozygous Axin2-mTurquoise2 mice (Tg/wt) and four homozygous Axin2-mTurquoise2 mice (Tg/Tg) from two independent experiments. Data bars represent median values. Error bars represent minimum and maximum values. * *p* < 0.033 and *** *p* < 0.001 (ANOVA test).

**Figure 3 cells-11-02679-f003:**
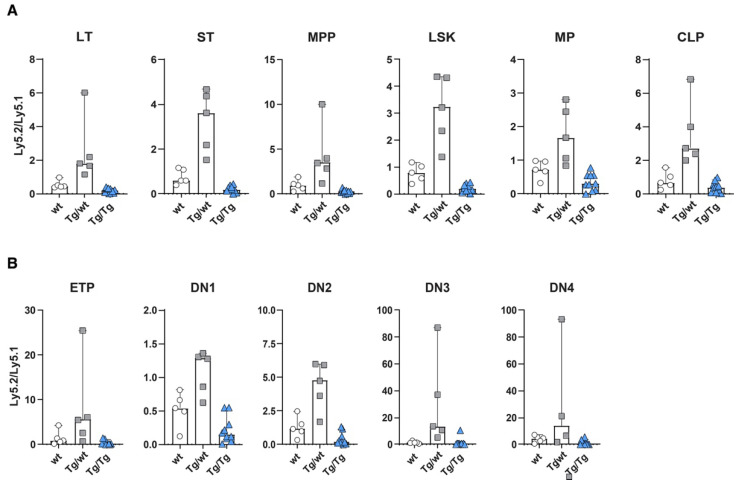
Wnt signaling dosage model alters HSC function and early T cell development. Repopulation efficiency analyzed in the bone marrow (**A**) and thymuses (**B**) of mice transplanted with Ly5.2 wild-type (wt), heterozygous Axin2-mTurquoise2 (Tg/wt), homozygous Axin2-mTurquoise2 (Tg/Tg) and Ly5.1 wild-type control HSCs, 6 weeks after transplantation. Data represent the Ly5.2-to-Ly5.1 ratio results for five wild-type control mice (wt), five heterozygous Axin2-mTurquoise2 mice (Tg/wt) and ten homozygous Axin2-mTurquoise2 mice (Tg/Tg) from two independent experiments. Error bars represent SEMs.

**Figure 4 cells-11-02679-f004:**
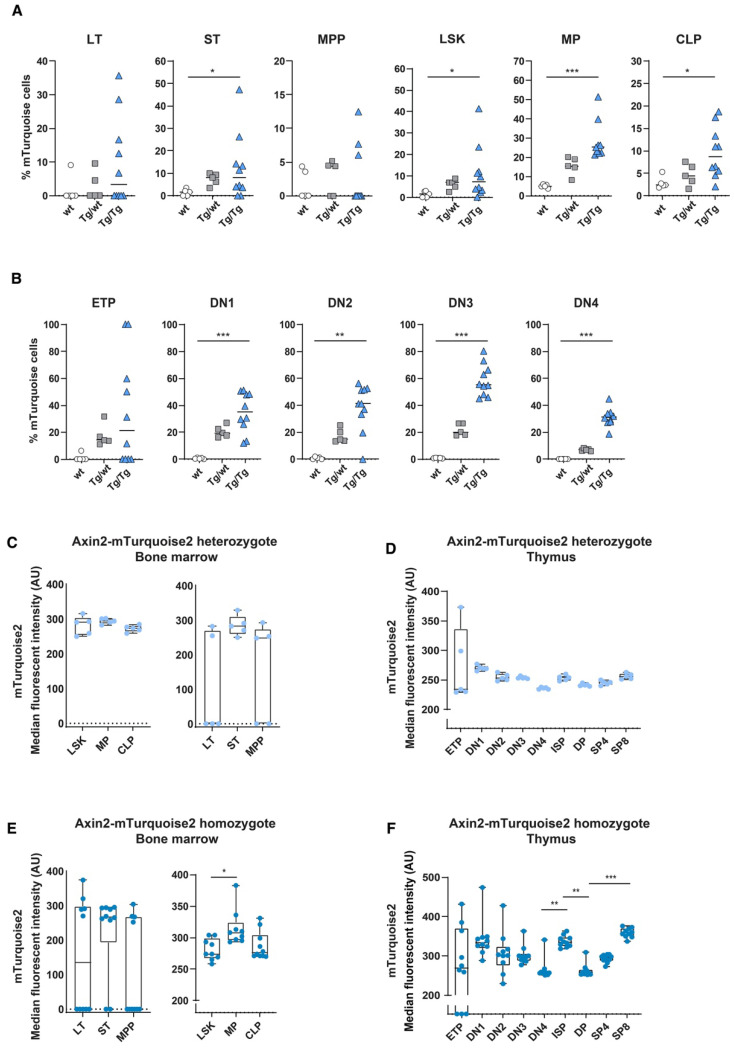
In vivo measurement of canonical Wnt signaling activity in hematopoietic stem cells and T cell development after competitive transplantation. Activation of the canonical Wnt signaling pathway was measured in the Axin2-mTurquoise2 test population (Ly5.2) fraction of the transplanted recipient mice by FACS. Quantification of the frequency of mTurquoise2 (**A**,**B**) for each subset in the bone marrow (**A**) and in the thymus (**B**). The test population (Ly5.2) fraction not carrying the reporter transgene (Axin2-mTurquoise2 wild type) was used to define the mTurquoise2 population. Data represent results for five wild-type control mice (wt), five heterozygous Axin2-mTurquoise2 mice (Tg/wt) and ten homozygous Axin2-mTurquoise2 mice (Tg/Tg) from two independent experiments. (**C**,**D**) Quantification of the median fluorescence intensity (MFI) of the mTurquoise2 populations of heterozygous Axin2-mTurquoise2 (Tg/wt) mice for each subset in the bone marrow (**C**) and in the thymus (**D**). (**E**,**F**) Quantification of the median fluorescence intensity (MFI) of the mTurquoise2 populations of homozygous Axin2-mTurquoise2 (Tg/Tg) mice for each subset in the bone marrow (**E**) and in the thymus (**F**). Data bars represent median values. * *p* < 0.033, ** *p* < 0.002 and *** *p* < 0.001 (ANOVA test).

**Figure 5 cells-11-02679-f005:**
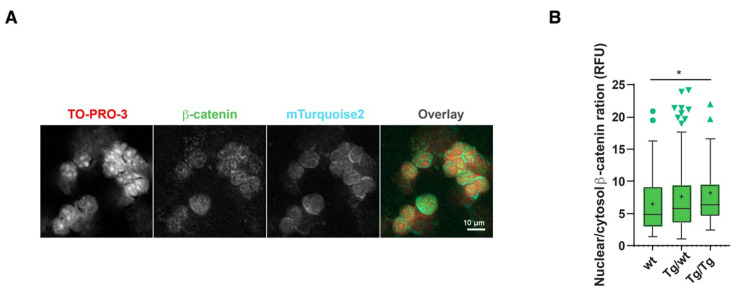
Axin2 allele deficiency leads to increased canonical Wnt signaling. Measurement of Axin2-mTurquoise2 and β-catenin expression by confocal imaging in thymocyte cytospin preparations. (**A**) Confocal grayscale and overlay representation of homozygote (Tg/Tg) Axin2-mTurquoise2 total thymocytes stained with nuclear TO-PRO-3 and total β-catenin AF568, as well as endogenous cytoplasmic mTurquoise2 expression. Images were taken at 40× with 1.5 zoom factor. (**B**) Boxplot ratio representation of nuclear (active) to cytoplasmic (inactive) total β-catenin AF568 intensity values from wild-type (wt), heterozygous (Tg/wt) and homozygous (Tg/Tg) Axin2-mTurquoise2 mice (50–70 cells per genotype). Data bars represent median values; protruding data points are outliers from the median boxplot; + signs represent mean values. Error bars represent Tukey whiskers. * *p* < 0.033 (Mann–Whitney U test).

## Data Availability

Not applicable.

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
