# Peer review of "Axin2/Conductin Is Required for Normal Haematopoiesis and T Lymphopoiesis"

_cells, 2022, doi:10.3390/cells11172679_

Round 1
Reviewer 1 Report
In this manuscript the authors use a newly generated mouse model to investigate the role of Wnt signaling in thymopoiesis and hematopoietic stem cells. They developed a mouse where the Axin2 gene was replaced with a reporter allele that generated a fluorescent signal to track cells expressing the gene, a target of canonical Wnt signaling. Overall, they demonstrate that T lymphopoiesis is impaired in homozygous mice carrying two reporter alleles, effectively blunting Wnt signaling. These data are compelling but several comments below should be considered and addressed to improve the manuscript:
1. Imaging data in Figure 1 are very difficult to see, even the labels are much too small to read. Is there additional description of the text that can explain these images?
2. It is not immediately clear why there is such variation in reporter gene expression in some populations (ST-HSCs; Figure 2).
3. The way the data is presented in Figure 3 is counterintuitive to the conclusions because the full Axin2 knockout has poor reconstitution, but the graphical display conveys the opposite. This should be modified for clarity.
4. Data in Figure 5 are confusing and not convincing. What are the images showing? Why are the box plots not aligning with the data points? What is being shown exactly?
Minor comments:
1. Read for typos; “Stinkingly” should be “strikingly”? Line 252
2. Figure legends are difficult to follow in some cases. Use semi-colons or commas to separate.
Author Response
We thank all three reviewers for carefully reading our manuscript and for their useful suggestions which have improved the paper.
- Imaging data in Figure 1 are very difficult to see, even the labels are much too small to read. Is there additional description of the text that can explain these images?
A: we have adapted figure 1D with larger labels as well as a description how to interpret the pictures. We also indicate (see point 4 by this reviewer) how to interpret these data in terms of cell-based mTurquoise2 measurements.
- It is not immediately clear why there is such variation in reporter gene expression in some populations (ST-HSCs; Figure 2).
A: The variation is caused by the relatively low number of stem cells in the gated data. We have added an explanation in the text.
- The way the data is presented in Figure 3 is counterintuitive to the conclusions because the full Axin2 knockout has poor reconstitution, but the graphical display conveys the opposite. This should be modified for clarity.
A:We thank the reviewer for this very useful suggestion. We have now switched from
Ly5.1/Ly5.2 to Ly5.2/Ly5.1, which makes the data interpretation more intuitive
- Data in Figure 5 are confusing and not convincing. What are the images showing? Why are the box plots not aligning with the data points? What is being shown exactly?
A:What we try to show here is the median of the ratios nuclear/cytoplasmic staining per condition, with the visible symbols showing outliers. Consulting microscopy experts, we were told that this is the most appropriate way to show the collective data from a large number of images. Given the large number of data points there is s a statistical significance between wt and Tg/Tg (the Axin2 KO).
Minor comments:
1. Read for typos; “Stinkingly” should be “strikingly”? Line 252
A:Changed
- Figure legends are difficult to follow in some cases. Use semi-colons or commas to separate.
A: Changed were appropriate
Reviewer 2 Report
Jolanda J.D. de Roo et al. introduced a mouse model to investigate the Wnt pathway in normal haematopoiesis and T lymphopoiesis, but the mouse model can not represent the Wnt activity, here are my comments:
Major:
1. The authors claimed that “mTurquoise2 activity equal to Wnt activity” in the transgenic mice, the reviewer do not think the mTurquoise2 activity can represent Wnt activity. The mTurquoise2 was under control of Axin2 promoter, it can only indicate the Axin2 promoter activity, how can it represent Wnt activity? The Wnt pathway contains many genes, the core element is the beta-Catenin protein, the only protein can represent Wnt pathway activity.
2. Did the groups have significant differences in Figure 3??
3. The ratio of beta-Catenin in nuclei and cytosol (Fig 5) is not a good method to indicate Wnt signaling activity. Since the nuclear accumulation of beta-Catenin is the key process of Wnt pathways activity, the authors should detect the nuclear beta-Catenin expression, or at least the expression in whole cells.
Minor:
1. Line 53, the last word should be beta.
Author Response
Major:
- The authors claimed that “mTurquoise2 activity equal to Wnt activity” in the transgenic mice, the reviewer do not think the mTurquoise2 activity can represent Wnt activity. The mTurquoise2 was under control of Axin2 promoter, it can only indicate the Axin2 promoter activity, how can it represent Wnt activity? The Wnt pathway contains many genes, the core element is the beta-Catenin protein, the only protein can represent Wnt pathway activity.
A:We respectfully disagree with the reviewer about usage of Axin2 expression as Wnt reporter. For many signaling pathways, the activity of such a pathway can be faithfully measured by using the expression of target genes induced by such a pathway. For many years now, Axin2 expression has fulfilled that role in the Wnt pathway. Indeed as the reviewer correctly points out, the levels of b-catenin protein also are important as potential readout of Wnt activity, but as we reviewed in ref 2 and 10 there are other ways to measure Wnt activity besides b-catenin levels;. Even for b-catenin it is best to measure the nuclear dephosphorylated b-catenin amounts, but other measures such as the Top-Fop reporter assays, or levels of Axin2 are also appropriate as extensively discussed in ref 2: Staal & Clevers Nature Rev Immunol. 5, 21-30 (2005). In order to facilitate measurements of Axin2 by flow cytometry and microscopy in intact cells (rather than measuring mRNA levels) the current Turquoise knockin in model was developed. (see ref 16 de Roo JJD, Breukel C, Chhatta AR, Linssen MM, Vloemans SA, Salvatori D, et al. Axin2-mTurquoise2: A novel reporter mouse model for the detection of canonical Wnt signalling. Genesis. 55, (2017).)
- Did the groups have significant differences in Figure 3??
A: There is too much spread in the in vivo results to reach statistical significance. But the trends are visible as indicated in the text.
- The ratio of beta-Catenin in nuclei and cytosol (Fig 5) is not a good method to indicate Wnt signaling activity. Since the nuclear accumulation of beta-Catenin is the key process of Wnt pathways activity, the authors should detect the nuclear beta-Catenin expression, or at least the expression in whole cells.
A: We have better explained why the ratios of nuclear to cytoplasmic b-catenin are a good way to measure Wnt signaling, especially in the setting of Axin2 deletion. As Axin2 is part of the destruction complex, its inactivation can result in alterations in b-catenin localization. We are trying to give a complete picture of the effects on b-catenin and Wnt activity this way.
Minor:
- Line 53, the last word should be beta.
A: Changed
Reviewer 3 Report
The authors have generated a mTurquoise2 fluorescent protein-based reporter transgene driven by the Axin2 promoter to enable monitoring of the WNT signaling pathway. This represents a significant improvement of the previous Axin2-LacZ reporter from the same laboratory. The paper shows data from experiments with heterozygous and homozygous transgenic mice and demonstrate the usefulness of this reporter allele for Wnt signaling studies. Also, the authors can conclude from their data that Axin2 plays a non-redundant role in regulating HSC differentiation and early T cell development. This is a well-written paper with very valuable information for the Wnt signaling community and in particular provides information on the new Axin2 reporter mouse for future users. There are only a few minor point that should be addressed:
- Fig. 1: The Axin2-mTurquoise2 heterozygote (Tg/0) could be called Tg/wt. The use of the “0” here is confusing, unless there is a specific reason.
- Panel D should be enlarged to reveal more detail.
- In vivo and in vitro should be in italics throughout the MS
- Line 228: correct “@-” for “beta”-galactosidase staining procedure
Line 252: correct “Stinkingly”, the ETP subpopulation… for “strikingly”
Author Response
- Fig. 1: The Axin2-mTurquoise2 heterozygote (Tg/0) could be called Tg/wt. The use of the “0” here is confusing, unless there is a specific reason. A: Changed
- Panel D should be enlarged to reveal more detail. A: Changed
- In vivo and in vitro should be in italics throughout the MS A: Changed
- Line 228: correct “@-” for “beta”-galactosidase staining procedure A: Changed
Line 252: correct “Stinkingly”, the ETP subpopulation… for “strikingly” A:Changed
Round 2
Reviewer 2 Report
Maybe the reviewer said "beta-Catenin protein, the only protein can represent Wnt pathway activity." is to restrict, the appropriate is "beta-Catenin protein is the best protein that can represent Wnt pathway activity.". The key process of Wnt pathway (canonical) is the accumulation of b-Catenin in nuclei and regulating the downstream genes' expressions. If the authors didn't present this kind of data, it is hard to convince the reviewer that this method (Axin2 promoter activity) can represent the activity of Wnt pathway.
As point to the third question, the key process of Wnt pathway (canonical) is the accumulation of b-Catenin in nuclei. The key point is the "accumulation in nuclei", which activates the Wnt pathway, not the ratio. The best situation to use ratio is the whole protein levels of b-Catenin is unchanged. If the protein level of b-Catenin in nuclei is decreased, but it decreased more in cytosol, using the authors' method (ratio) the conclusion is "the Wnt activity is increased", actually the activity of Wnt is decreased since the b-Catenin in nuclei is less. So the authors didn't convince me that their methods are appropriate to represent the Wnt activity.
Author Response
As we now explain extensively (newly added paragraph in discussion; pg 14 lines 506-516) there are various ways to measure Wnt signaling, Axin2 expression being one of the most widely accepted and used readouts in the field. We agree that b-catenin levels per se do not say everything that is why we measured activated b-catenin, the dephosphorylated and signaling competent form of b-catenin. The reviewer may have missed this point, which we now explain more thoroughly.